# Faster Deep Reinforcement Learning with Slower Online Network

**Kavosh Asadi**
Amazon Web Services

**Rasool Fakoor**
Amazon Web Services

**Omer Gottesman**
Brown University

**Taesup Kim**
Seoul National University

**Michael L. Littman**
Brown University

**Alexander J. Smola**
Amazon Web Services

## Abstract

Deep reinforcement learning algorithms often use two networks for value function optimization: an online network, and a target network that tracks the online network with some delay. Using two separate networks enables the agent to hedge against issues that arise when performing bootstrapping. In this paper we endow two popular deep reinforcement learning algorithms, namely DQN and Rainbow, with updates that incentivize the online network to remain in the proximity of the target network. This improves the robustness of deep reinforcement learning in presence of noisy updates. The resultant agents, called DQN Pro and Rainbow Pro, exhibit significant performance improvements over their original counterparts on the Atari benchmark demonstrating the effectiveness of this simple idea in deep reinforcement learning. The code for our paper is available here: Github.com/amazon-research/fast-rl-with-slow-updates.

## 1 Introduction

An important competency of reinforcement-learning (RL) agents is learning in environments with large state spaces like those found in robotics (Kober et al., 2013), dialog systems (Williams et al., 2017), and games (Tesauro, 1994; Silver et al., 2017). Recent breakthroughs in deep RL have demonstrated that simple approaches such as Q-learning (Watkins & Dayan, 1992) can surpass human-level performance in challenging environments when equipped with deep neural networks for function approximation (Mnih et al., 2015).

Two components of a gradient-based deep RL agent are its objective function and optimization procedure. The optimization procedure takes estimates of the gradient of the objective with respect to network parameters and updates the parameters accordingly. In DQN (Mnih et al., 2015), for example, the objective function is the empirical expectation of the temporal difference (TD) error (Sutton, 1988) on a buffered set of environmental interactions (Lin, 1992), and variants of stochastic gradient descent are employed to best minimize this objective function.

A fundamental difficulty in this context stems from the use of bootstrapping. Here, bootstrapping refers to the dependence of the target of updates on the parameters of the neural network, which is itself continuously updated during training. Employing bootstrapping in RL stands in contrast to supervised-learning techniques and Monte-Carlo RL (Sutton & Barto, 2018), where the target of our gradient updates does not depend on the parameters of the neural network.

Mnih et al. (2015) proposed a simple approach to hedging against issues that arise when using bootstrapping, namely to use a *target network* in value-function optimization. The target network is updated periodically, and tracks the online network with some delay. While this modification

constituted a major step towards combating misbehavior in Q-learning (Lee & He, 2019; Kim et al., 2019; Zhang et al., 2021), optimization instability is still prevalent (van Hasselt et al., 2018).

Our primary contribution is to endow DQN and Rainbow (Hessel et al., 2018) with a term that ensures the parameters of the online-network component remain in the proximity of the parameters of the target network. Our theoretical and empirical results show that our simple proximal updates can remarkably increase robustness to noise without incurring additional computational or memory costs. In particular, we present comprehensive experiments on the Atari benchmark (Bellemare et al., 2013) where proximal updates yield significant improvements, thus revealing the benefits of using this simple technique for deep RL.

## 2 Background and Notation

RL is the study of the interaction between an environment and an agent that learns to maximize reward through experience. The Markov Decision Process (Puterman, 1994), or MDP, is used to mathematically define the RL problem. An MDP is specified by the tuple $\langle \mathcal{S}, \mathcal{A}, \mathcal{R}, \mathcal{P}, \gamma \rangle$, where $\mathcal{S}$ is the set of states and $\mathcal{A}$ is the set of actions. The functions $\mathcal{R} : \mathcal{S} \times \mathcal{A} \to \mathbb{R}$ and $\mathcal{P} : \mathcal{S} \times \mathcal{A} \times \mathcal{S} \to [0, 1]$ denote the reward and transition dynamics of the MDP. Finally, a discounting factor $\gamma$ is used to formalize the intuition that short-term rewards are more valuable than those received later.

The goal in the RL problem is to learn a policy, a mapping from states to a probability distribution over actions, $\pi : \mathcal{S} \to \mathcal{P}(\mathcal{A})$, that obtains high sums of future discounted rewards. An important concept in RL is the state value function. Formally, it denotes the expected discounted sum of future rewards when committing to a policy $\pi$ in a state $s$: $v^\pi(s) := \mathbb{E}\big[\sum_{t=0}^{\infty} \gamma^t R_t \big| S_0 = s, \pi \big]$. We define the Bellman operator $\mathcal{T}^\pi$ as follows:

$$\big[\mathcal{T}^\pi v\big](s) := \sum_{a \in \mathcal{A}} \pi(a \mid s)\big(\mathcal{R}(s, a) + \sum_{s' \in \mathcal{S}} \gamma \, \mathcal{P}(s, a, s')v(s')\big),$$

which we can write compactly as: $\mathcal{T}^\pi v := R^\pi + \gamma P^\pi v$, where $\big[R^\pi\big](s) = \sum_{a \in \mathcal{A}} \pi(a|s)\mathcal{R}(s, a)$ and $\big[P^\pi v\big](s) = \sum_{a \in \mathcal{A}} \pi(a \mid s) \sum_{s' \in \mathcal{S}} \mathcal{P}(s, a, s')v(s')$. We also denote: $(\mathcal{T}^\pi)^n v := \underbrace{\mathcal{T}^\pi \cdots \mathcal{T}^\pi}_{n \text{ compositions}} v$.

Notice that $v^\pi$ is the unique fixed-point of $(\mathcal{T}^\pi)^n$ for all natural numbers $n$, meaning that $v^\pi = (\mathcal{T}^\pi)^n v^\pi$, for all $n$. Define $v^\star$ as the optimal value of a state, namely: $v^\star(s) := \max_\pi v^\pi(s)$, and $\pi^\star$ as a policy that achieves $v^\star(s)$ for all states. We define the Bellman Optimality Operator $\mathcal{T}^\star$:

$$\big[\mathcal{T}^\star v\big](s) := \max_{a \in \mathcal{A}} \mathcal{R}(s, a) + \sum_{s' \in \mathcal{S}} \gamma \, \mathcal{P}(s, a, s')v(s'),$$

whose fixed point is $v^\star$. These operators are at the heart of many planning and RL algorithms including Value Iteration (Bellman, 1957) and Policy Iteration (Howard, 1960).

## 3 Proximal Bellman Operator

In this section, we introduce a new class of Bellman operators that ensure that the next iterate in planning and RL remain in the vicinity of the previous iterate. To this end, we define the Bregman Divergence generated by a convex function $f$:

$$D_f(v', v) := f(v') - f(v) - \langle \nabla f(v), v' - v \rangle.$$

Examples include the $l_p$ norm generated by $f(v) = \frac{1}{2}\|v\|_p^2$ and the Mahalanobis Distance generated by $f(v) = \frac{1}{2}\langle v, Qv \rangle$ for a positive semi-definite matrix $Q$.

We now define the Proximal Bellman Operator $(\mathcal{T}_{c,f}^\pi)^n$:

$$(\mathcal{T}_{c,f}^\pi)^n v := \arg\min_{v'} \|v' - (\mathcal{T}^\pi)^n v\|_2^2 + \frac{1}{c} D_f(v', v), \tag{1}$$

where $c \in (0, \infty)$. Intuitively, this operator encourages the next iterate to be in the proximity of the previous iterate, while also having a small difference relative to the point recommended by the

original Bellman Operator. The parameter $c$ could, therefore, be thought of as a knob that controls the degree of gravitation towards the previous iterate.

Our goal is to understand the behavior of Proximal Bellman Operator when used in conjunction with the Modified Policy Iteration (MPI) algorithm (Puterman, 1994; Scherrer et al., 2015). Define $\mathcal{G}v$ as the greedy policy with respect to $v$. At a certain iteration $k$, Proximal Modified Policy Iteration (PMPI) proceeds as follows:

$$\pi_k \quad \leftarrow \quad \mathcal{G}v_{k-1} \, , \tag{2}$$
$$v_k \quad \leftarrow \quad (\mathcal{T}_{c,f}^{\pi_k})^n v_{k-1} \, . \tag{3}$$

The pair of updates above generalize existing algorithms. Notably, with $c \to \infty$ and general $n$ we get MPI, with $c \to \infty$ and $n = 1$ the algorithm reduces to Value Iteration, and with $c \to \infty$ and $n = \infty$ we have a reduction to Policy Iteration. For finite $c$, the two extremes of $n$, namely $n = 1$ and $n = \infty$, could be thought of as the proximal versions of Value Iteration and Policy Iteration, respectively.

To analyze this approach, it is first natural to ask if each iteration of PMPI could be thought of as a contraction so we can get sound and convergent behavior in planning and learning. For $n > 1$, Scherrer et al. (2015) constructed a contrived MDP demonstrating that one iteration of MPI can unfortunately expand. As PMPI is just a generalization of MPI, the same example from Scherrer et al. (2015) shows that PMPI can expand. In the case of $n = 1$, we can rewrite the pair of equations (2) and (3) in a single update as follows: $v_k \leftarrow \mathcal{T}_{c,f}^\star v_{k-1}$ . When $c \to \infty$, standard proofs can be employed to show that the operator is a contraction (Littman & Szepesvári, 1996). We now show that $\mathcal{T}_{c,f}^\star$ is a contraction for finite values of $c$. See our appendix for proofs.

**Theorem 1.** *The Proximal Bellman Optimality Operator $\mathcal{T}_{c,f}^\star$ is a contraction with fixed point $v^\star$.*

Therefore, we get convergent behavior when using $\mathcal{T}_{c,f}^\star$ in planning and RL. The addition of the proximal term is fortunately not changing the fixed point, thus not negatively affecting the final solution. This could be thought of as a form of regularization that vanishes in the limit; the algorithm converges to $v^\star$ even without decaying $1/c$.

Going back to the general $n \geq 1$ case, we cannot show contraction, but following previous work (Bertsekas & Tsitsiklis, 1996; Scherrer et al., 2015), we study error propagation in PMPI in presence of additive noise where we get a noisy sample of the original Bellman Operator $(\mathcal{T}^{\pi_k})^n v_{k-1} + \epsilon_k$. The noise can stem from a variety of reasons, such as approximation or estimation error. For simplicity, we restrict the analysis to $D_f(v', v) = ||v' - v||_2^2$, so we rewrite update (3) as:

$$v_k \leftarrow \arg\min_{v'} ||v' - ((\mathcal{T}^{\pi_k})^n v_{k-1} + \epsilon_k)||_2^2 + \frac{1}{c} \, ||v' - v_{k-1}||_2^2$$

which can further be simplified to:

$$v_k \leftarrow \underbrace{(1 - \beta)(\mathcal{T}^{\pi_k})^n v_{k-1} + \beta v_{k-1}}_{:=(\mathcal{T}_\beta^{\pi_k})^n v_{k-1}} + (1 - \beta)\epsilon_k,$$

where $\beta = \frac{1}{1+c}$. This operator is a generalization of the operator proposed by Smirnova & Dohmatob (2020) who focused on the case of $n = 1$. To build some intuition, notice that the update is multiplying error $\epsilon_k$ by a term that is smaller than one, thus better hedging against large noise. While the update may slow progress when there is no noise, it is entirely conceivable that for large enough values of $\epsilon_k$, it is better to use non-zero $\beta$ values. In the following theorem we formalize this intuition. Our result leans on the theory provided by Scherrer et al. (2015) and could be thought of as a generalization of their theorem for non-zero $\beta$ values.

**Theorem 2.** *Consider the PMPI algorithm specified by:*

$$\pi_k \quad \leftarrow \quad \mathcal{G}_{\epsilon_k'} v_{k-1} \, , \tag{4}$$
$$v_k \quad \leftarrow \quad (\mathcal{T}_\beta^{\pi_k})^n v_{k-1} + (1 - \beta)\epsilon_k \, . \tag{5}$$

*Define the Bellman residual $b_k := v_k - \mathcal{T}^{\pi_{k+1}} v_k$, and error terms $x_k := (I - \gamma P^{\pi_k})\epsilon_k$ and $y_k := \gamma P^{\pi^*} \epsilon_k$. After $k$ steps:*

$$v^* - v^{\pi_k} = \underbrace{v^{\pi^*} - (\mathcal{T}_\beta^{\pi_{k+1}})^n v_k}_{d_k} + \underbrace{(\mathcal{T}_\beta^{\pi_{k+1}})^n v_k - v_{\pi_k}}_{s_k}$$

- *where $d_k \leq \gamma P^{\pi^*} d_{k-1} - \left((1-\beta)y_{k-1} + \beta b_{k-1}\right) + (1-\beta)\sum_{j=1}^{n-1}(\gamma P^{\pi_k})^j b_{k-1} + \epsilon'_k$*

- $s_k \leq \left((1-\beta)(\gamma P^{\pi_k})^n + \beta I\right)(I - \gamma P^{\pi_k})^{-1} b_{k-1}$

- $b_k \leq \left((1-\beta)(\gamma P^{\pi_k})^n + \beta I\right)b_{k-1} + (1-\beta)x_k + \epsilon'_{k+1}$

The bound provides intuition as to how the proximal Bellman Operator can accelerate convergence in the presence of high noise. For simplicity, we will only analyze the effect of the $\epsilon$ noise term, and ignore the $\epsilon'$ term. We first look at the Bellman residual, $b_k$. Given the Bellman residual in iteration $k-1$, $b_{k-1}$, the only influence of the noise term $\epsilon_k$ on $b_k$ is through the $(1-\beta)x_k$, term, and we see that $b_k$ decreases linearly with larger $\beta$.

The analysis of $s_k$ is slightly more involved but follows similar logic. The bound for $s_k$ can be decomposed into a term proportional to $b_{k-1}$ and a term proportional to $\beta b_{k-1}$, where both are multiplied with positive semi-definite matrices. Since $b_{k-1}$ itself linearly decreases with $\beta$, we conclude that larger $\beta$ decreases the bound quadratically.

The effect of $\beta$ on the bound for $d_k$ is more complex. The terms $\beta y_{k-1}$ and $\sum_{j=1}^{n-1}(\gamma P^{\pi_k})^j b_{k-1}$ introduce a linear decrease of the bound on $d_k$ with $\beta$, while the term $\beta(I - \sum_{j=1}^{n-1}(\gamma P^{\pi_k})^j)b_{k-1}$ introduces a quadratic dependence whose curvature depends on $I - \sum_{j=1}^{n-1}(\gamma P_{\pi_k})^j$. This complex dependence on $\beta$ highlights the trade-off between noise reduction and magnitude of updates. To understand this trade-off better, we examine two extreme cases for the magnitude on the noise. When the noise is very large, we may set $\beta = 1$, equivalent to an infinitely strong proximal term. It is easy to see that for $\beta = 1$, the values of $d_k$ and $s_k$ remain unchanged, which is preferable to the increase they would suffer in the presence of very large noise. On the other extreme, when no noise is present, the $x_k$ and $y_k$ terms in Theorem 2 vanish, and the bounds on $d_k$ and $s_k$ can be minimized by setting $\beta = 0$, i.e. without noise the proximal term should not be used and the original Bellman update performed. Intermediate noise magnitudes thus require a value of $\beta$ that balances the noise reduction and update size.

## 4  Deep Q-Network with Proximal Updates

We now endow DQN-style algorithms with proximal updates. Let $\langle s, a, r, s' \rangle$ denote a buffered tuple of interaction. Define the following objective function:

$$h(\theta, w) := \widehat{\mathbb{E}}_{\langle s,a,r,s' \rangle}\left[\left(r + \gamma \max_{a'} \widehat{Q}(s', a'; \theta) - \widehat{Q}(s, a; w)\right)^2\right]. \tag{6}$$

Our proximal update is defined as follows:

$$w_{t+1} \leftarrow \arg\min_w h(w_t, w) + \frac{1}{2\tilde{c}}\|w - w_t\|_2^2 . \tag{7}$$

This algorithm closely resembles the standard proximal-point algorithm (Rockafellar, 1976; Parikh & Boyd, 2014) with the important caveat that the function $h$ is now taking two vectors as input. At each iteration, we hold the first input constant while optimizing over the second input.

In the optimization literature, the proximal-point algorithm is well-studied in contexts where an analytical solution to (7) is available. With deep learning no closed-form solution exists, so we approximately solve (7) by taking a fixed number of descent steps using stochastic gradients. Specifically, starting each iteration with $w = w_t$, we perform multiple $w$ updates $w \leftarrow w - \alpha\left(\nabla_2 h(w_t, w) + \frac{1}{\tilde{c}}(w - w_t)\right)$. We end the iteration by setting $w_{t+1} \leftarrow w$. To make a connection to standard deep RL, the online weights $w$ could be thought of as the weights we maintain in the interim to solve (7) due to lack of a closed-form solution. Also, what is commonly referred to as the target network could better be thought of as just the previous iterate in the above proximal-point algorithm.

Observe that the update can be written as: $w \leftarrow \left(1 - (\alpha/\tilde{c})\right) \cdot w + (\alpha/\tilde{c}) \cdot w_t - \alpha\nabla_2 h(w_t, w) $. Notice the intuitively appealing form: we first compute a convex combination of $w_t$ and $w$, based on the hyper-parameters $\alpha$ and $\tilde{c}$, then add the gradient term to arrive at the next iterate of $w$. If $w_t$ and $w$ are close, the convex combination is close to $w$ itself and so this DQN with proximal update

(*DQN Pro*) would behave similarly to the original DQN. However, when $w$ strays too far from $w_t$, taking the convex combination ensures that $w$ gravitates towards the previous iterate $w_t$. The gradient signal from minimizing the squared TD error (6) should then be strong enough to cancel this default gravitation towards $w_t$. The update includes standard DQN as a special case when $\tilde{c} \to \infty$. The pseudo-code for DQN is presented in the Appendix. The difference between DQN and DQN Pro is minimal (shown in gray), and corresponds with a few lines of code in our implementation.

## 5 Experiments

In this section, we empirically investigate the effectiveness of proximal updates in planning and reinforcement-learning algorithms. We begin by conducting experiments with PMPI in the context of approximate planning, and then move to large-scale RL experiments in Atari.

### 5.1 PMPI Experiments

We now focus on understanding the empirical impact of adding the proximal term on the performance of approximate PMPI. To this end, we use the pair of update equations:

$$
\begin{aligned}
\pi_k &\leftarrow \mathcal{G}v_{k-1} \,, \\
v_k &\leftarrow (1-\beta)\big((\mathcal{T}^{\pi_k})^n v_{k-1} + \epsilon_k\big) + \beta v_{k-1} \,.
\end{aligned}
$$

For this experiment, we chose the toy $8 \times 8$ Frozen Lake environment from Open AI Gym (Brockman et al., 2016), where the transition and reward model of the environment is available to the planner. Using a small environment allows us to understand the impact of the proximal term in the simplest and most clear setting. Note also that we arranged the experiment so that the policy greedification step $\mathcal{G}v_{k-1}\ \forall k$ is error-free, so we can solely focus on the interplay between the proximal term and the error caused by imperfect policy evaluation.

We applied 100 iterations of PMPI, then measured the quality of the resultant policy $\pi := \pi_{100}$ as defined by the distance between its true value and that of the optimal policy, namely $\|V^\star - V^\pi\|_\infty$. We repeated the experiment with different magnitudes of error, as well as different values of the $\beta$ parameter.

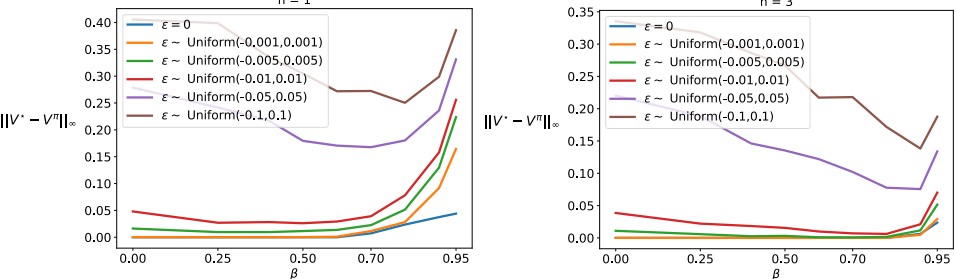

Figure 1: Performance of approximate PMPI has a U-shaped dependence on the parameter $\beta = \frac{1}{1+c}$. Results are averaged over 30 random seeds with $n = 1$ (**left**) and $n = 3$ (**right**).

From Figure 1, it is clear that the final performance exhibits a U-shape with respect to the parameter $\beta$. It is also noticable that the best-performing $\beta$ is shifting to the right side (larger values) as we increase the magnitude of noise. This trend makes sense, and is consistent with what is predicted by Theorem 2: As the noise level rises, we have more incentive to use larger (but not too large) $\beta$ values to hedge against it.

### 5.2 Atari Experiments

In this section, we evaluate the proximal (or Pro) agents relative to their original DQN-style counterparts on the Atari benchmark (Bellemare et al., 2013), and show that endowing the agent with the proximal term can lead into significant improvements in the interim as well as in the final performance. We next investigate the utility of our proposed proximal term through further experiments. Please see the Appendix for a complete description of our experimental pipeline.

### 5.2.1 Setup

We used 55 Atari games (Bellemare et al., 2013) to conduct our experimental evaluations. Following Machado et al. (2018) and Castro et al. (2018), we used sticky actions to inject stochasticity into the otherwise deterministic Atari emulator.

Our training and evaluation protocols and the hyper-parameter settings follow those of the Dopamine baseline (Castro et al., 2018). To report performance, we measured the undiscounted sum of rewards obtained by the learned policy during evaluation. We further report the learning curve for all experiments averaged across 5 random seeds. We reiterate that we used the exact same hyper-parameters for all agents to ensure a sound comparison.

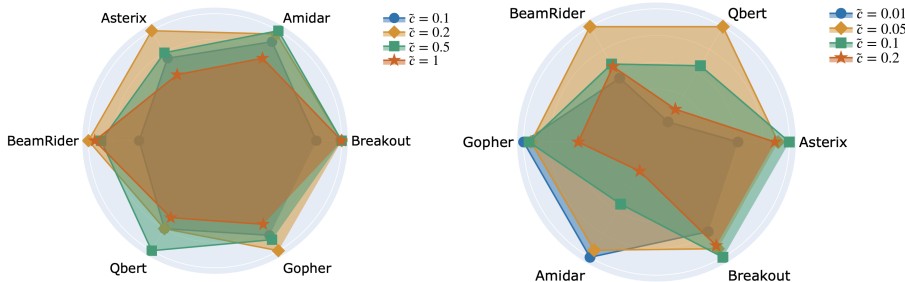

Figure 2: A minimal hyper-parameter tuning for $\tilde{c}$ in DQN Pro **(left)** and Rainbow Pro **(right)**.

Our Pro agents have a single additional hyper-parameter $\tilde{c}$. We did a minimal random search on 6 games to tune $\tilde{c}$. Figure 2 visualizes the performance of Pro agents as a function of $\tilde{c}$. In light of this result, we set $\tilde{c} = 0.2$ for DQN Pro and $\tilde{c} = 0.05$ for Rainbow Pro. We used these values of $\tilde{c}$ for all 55 games, and note that we performed no further hyper-parameter tuning at all.

### 5.2.2 Results

The first question is whether endowing the DQN agent with the proximal term can yield significant improvements over the original DQN. Figure 3 (top) shows a comparison between DQN and DQN Pro in terms of the final performance. In particular, following standard practice (Wang et al., 2016; Dabney et al., 2018; van Seijen et al., 2019), for each game we compute:

$$\frac{\text{Score}_{\text{DQN Pro}} - \text{Score}_{\text{DQN}}}{\max(\text{Score}_{\text{DQN}}, \text{Score}_{\text{Human}}) - \text{Score}_{\text{Random}}}.$$

Bars shown in red indicate the games in which we observed better final performance for DQN Pro relative to DQN, and bars in blue indicate the opposite. The height of a bar denotes the magnitude of this improvement for the corresponding benchmark; notice that the y-axis is scaled logarithmically. We took human and random scores from previous work (Nair et al., 2015; Dabney et al., 2018). It is clear that DQN Pro dramatically improves upon DQN. We defer to the Appendix for full learning curves on all games tested.

Can we fruitfully combine the proximal term with some of the existing algorithmic improvements in DQN? To answer this question, we build on the Rainbow algorithm of Hessel et al. (2018) who successfully combined numerous important algorithmic ideas in the value-based RL literature. We present this result in Figure 3 (bottom). Observe that the overall trend is for Rainbow Pro to yield large performance improvements over Rainbow.

Additionally, we measured the performance of our agents relative to human players. To this end, and again following previous work (Wang et al., 2016; Dabney et al., 2018; van Seijen et al., 2019), for each agent we compute the human-normalized score:

$$\frac{\text{Score}_{\text{Agent}} - \text{Score}_{\text{Random}}}{\text{Score}_{\text{Human}} - \text{Score}_{\text{Random}}}.$$

In Figure 4 (left), we show the median of this score for all agents, which Wang et al. (2016) and Hessel et al. (2018) argued is a sensible quantity to track. We also show per-game learning curves with standard error in the Appendix.

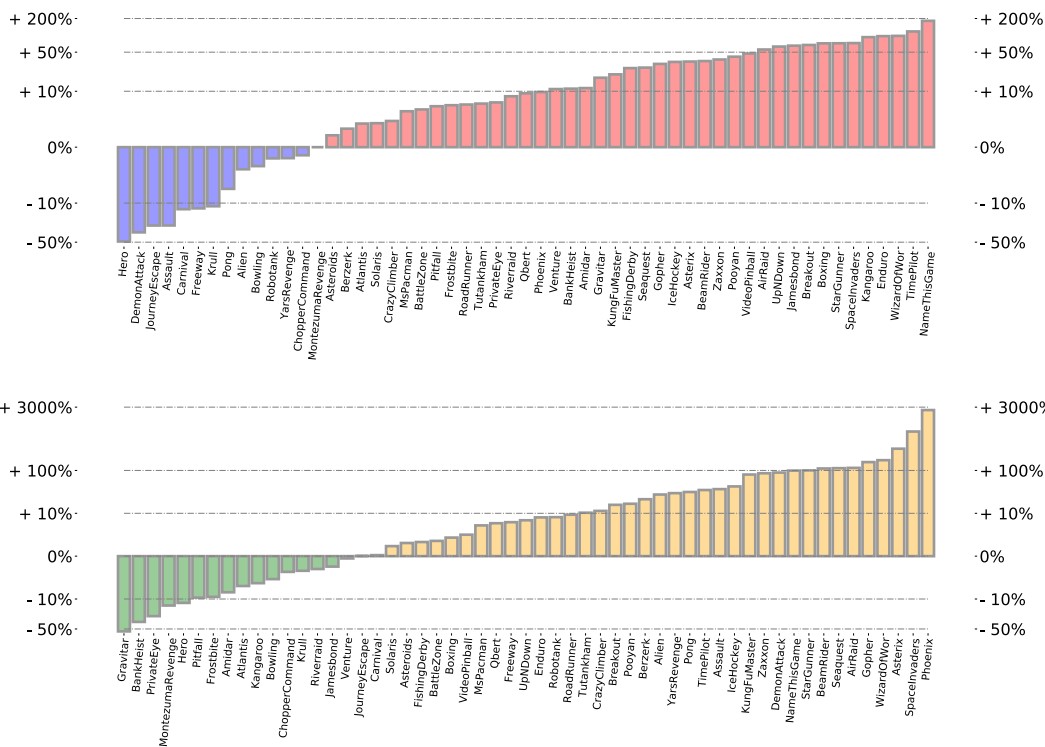

Figure 3: Final gain for DQN Pro over DQN **(top)**, and Rainbow Pro over Rainbow **(bottom)**, averaged over 5 seeds. DQN Pro and Rainbow Pro significantly outperform their original counterparts.

We make two key observations from this figure. First, the very basic DQN Pro agent is capable of achieving human-level performance (1.0 on the y-axis) after 120 million frames. Second, the Rainbow Pro agent achieves 220 percent human-normalized score after only 120 million frames.

### 5.2.3 Additional Experiments

Our purpose in endowing the agent with the proximal term was to keep the online network in the vicinity of the target network, so it would be natural to ask if this desirable property can manifest itself in practice when using the proximal term. In Figure 4, we answer this question affirmatively by plotting the magnitude of the update to the target network during synchronization. Notice that we periodically synchronize online and target networks, so the proximity of the online and target network should manifest itself in a low distance between two consecutive target networks. Indeed, the results demonstrate the success of the proximal term in terms of obtaining the desired proximity of online and target networks.

While using the proximal term leads to significant improvements, one may still wonder if the advantage of DQN Pro over DQN is merely stemming from a poorly-chosen *period* hyper-parameter in the original DQN, as opposed to a truly more stable optimization in DQN Pro. To refute this hypothesis, we ran DQN with various settings of the *period* hyper-parameter $\{2000, 4000, 8000, 12000\}$. This set included the default value of the hyper-parameter (8000) from the original paper (Mnih et al., 2015), but also covered a wider set of settings.

Additionally, we tried an alternative update strategy for the target network, referred to as Polyak averaging, which was popularized in the context of continuous-action RL (Lillicrap et al., 2015): $\theta \leftarrow \tau w + (1 - \tau)\theta$. For this update strategy, too, we tried different settings of the $\tau$ hyper-parameter, namely $\{0.05, 0.005, 0.0005\}$, which includes the value 0.005 used in numerous papers (Lillicrap et al., 2015; Fujimoto et al., 2018; Asadi et al., 2021).

Figure 5 presents a comparison between DQN Pro and DQN with periodic and Polyak target updates for various hyper-parameter settings of *period* and $\tau$. It is clear that DQN Pro is consistently

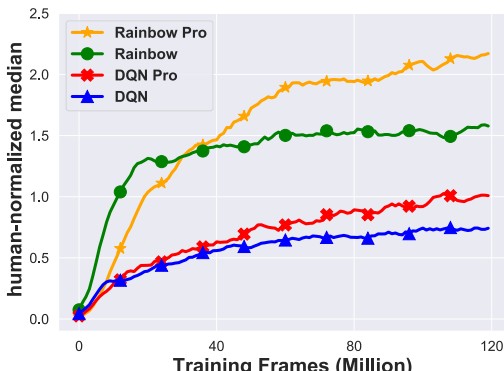
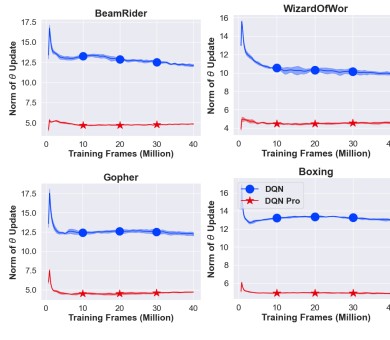

Figure 4: **(left)**: Human-normalized median performance for DQN, Rainbow, DQN Pro, and Rainbow Pro on 55 Atari games. Results are averaged over 5 independent seeds. Our agents, Rainbow Pro (yellow) and DQN Pro (red) outperform their original counterparts Rainbow (green) and DQN (blue). **(right)**: Using the proximal term reduces the magnitude of target network updates.

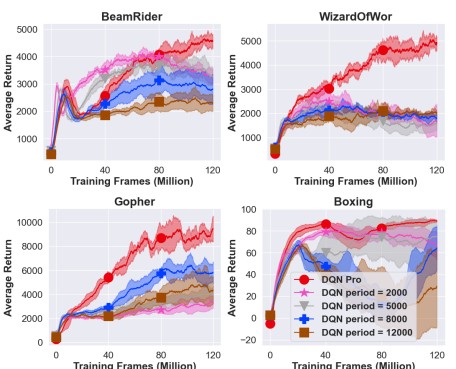
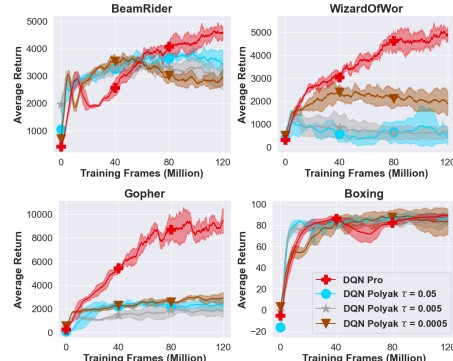

Figure 5: A comparison between DQN Pro and DQN with periodic **(left)** and Polyak **(right)** updates.

outperforming the two alternatives regardless of the specific values of *period* and $\tau$, thus clearly demonstrating that the improvement is stemming from a more stable optimization procedure leading to a better interplay between the two networks.

Finally, an alternative approach to ensuring lower distance between the online and the target network is to anneal the step size based on the number of updates performed on the online network since the last online-target synchronization. In this case we performed this experiment in 4 games where we knew proximal updates provide improvements based on our DQN Pro versus DQN resulst in Figure 3. In this case we linearly decreased the step size from the original DQN learning rate $\alpha$ to $\alpha' \ll \alpha$ where we tuned $\alpha'$ using random search. Annealing indeed improves DQN, but DQN Pro outperforms the improved version of DQN. Our intuition is that Pro agents only perform small updates when the target network is far from the online network, but naively decaying the learning rate can harm progress when the two networks are in vicinity of each other.

## 6 Discussion

In our experience using proximal updates in the parameter space were far superior than proximal updates in the value space. We believe this is because the parameter-space definition can enforce the proximity globally, while in the value space one can only hope to obtain proximity locally and on a batch of samples. One may hope to use natural gradients to enforce value-space proximity in a more principled way, but doing so usually requires significantly more computational resources Knight &

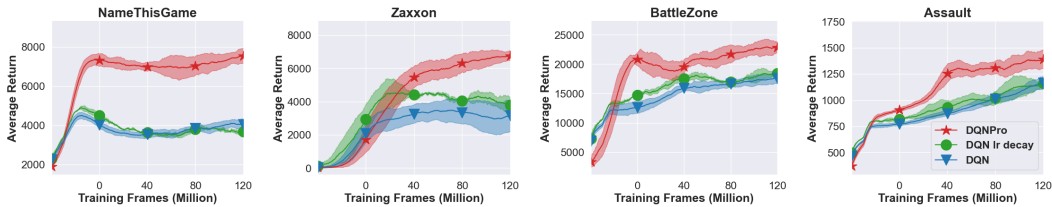

Figure 6: Learning-rate decay does not bridge the gap between DQN and DQN Pro.

Lerner (2018). This is in contrast to our proximal updates which add negligible computational cost in the simple form of taking a dimension-wise weighted average of two weight vectors.

In addition, for a smooth (Lipschitz) $Q$ function, performing parameter-space regularization guarantees function-space regularization. Concretely: $\forall s, \forall a \, |Q(s, a; \theta) - Q(s, a; \theta')| \leq L||\theta - \theta'||$, where $L$ is the Lipschitz constant of $Q$. Moreover, deep networks are Lipschitz (Neyshabur et al., 2015; Asadi et al., 2018), because they are constructed using compositions of Lipschitz functions (such as ReLU, convolutions, etc) and that composition of Lipschitz functions is Lipschitz. So performing value-space updates may be an overkill. Lipschitz property of deep networks has successfully been leveraged in other contexts, such as in generative adversarial training Arjovsky et al. (2017).

A key selling point of our result is simplicity, because simple results are easy to understand, implement, and reproduce. We obtained significant performance improvements by adding just a few lines of codes to the publicly available implementations of DQN and Rainbow Castro et al. (2018).

## 7 Related Work

The introduction of proximal operators could be traced back to the seminal work of Moreau (1962, 1965), Martinet (1970) and Rockafellar (1976), and the use of the proximal operators has since expanded into many areas of science such as signal processing (Combettes & Pesquet, 2009), statistics and machine learning (Beck & Teboulle, 2009; Polson et al., 2015; Reddi et al., 2015), and convex optimization (Parikh & Boyd, 2014; Bertsekas, 2011b;a).

In the context of RL, Mahadevan et al. (2014) introduced a proximal theory for deriving convergent off-policy algorithms with linear function approximation. One intriguing characteristic of their work is that they perform updates in primal-dual space, a property that was leveraged in sample complexity analysis (Liu et al., 2020) for the proximal counterparts of the gradient temporal-difference algorithm (Sutton et al., 2008). Proximal operators also have appeared in the deep RL literature. For instance, Fakoor et al. (2020b) used proximal operators for meta learning, and Maggipinto et al. (2020) improved TD3 (Fujimoto et al., 2018) by employing a stochastic proximal-point interpretation.

The effect of the proximal term in our work is reminiscent of the use of trust regions in policy-gradient algorithms (Schulman et al., 2015, 2017; Wang et al., 2019; Fakoor et al., 2020a; Tomar et al., 2021). However, three factors differentiate our work: we define the proximal term using the value function, not the policy, we enforce the proximal term in the parameter space, as opposed to the function space, and we use the target network as the previous iterate in our proximal definition.

## 8 Conclusion and Future work

We showed a clear advantage for using proximal terms to perform slower but more effective updates in approximate planning and reinforcement learning. Our results demonstrated that proximal updates lead to more robustness with respect to noise. Several improvements to proximal methods exist, such as the acceleration algorithm (Nesterov, 1983; Li & Lin, 2015), as well as using other proximal terms (Combettes & Pesquet, 2009), which we leave for future work.

## 9 Acknowledgment

We thank Lihong Li, Pratik Chaudhari, and Shoham Sabach for their valuable insights in different stages of this work.

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
