# 10  Appendix

## 10.1  Pseudo-code for DQN Pro

Below, we present the pseudo-code for DQN Pro. Notice that the difference between DQN and DQN Pro is minimal (highlighted in gray).

---

**Algorithm 1** DQN with Proximal Iteration (DQN Pro)

---

1: Initialize $\theta$, N, *period*, replay buffer $\mathcal{D}, \alpha$, and $\tilde{c}$
2: $s \leftarrow$ env.reset(), $w \leftarrow \theta$, numUpdates $\leftarrow 0$
3: **repeat**
4:     $a \sim \epsilon$-greedy$\big(Q(s, \cdot; w)\big)$
5:     $s', r \leftarrow$ env.step$(s, a)$
6:     add $\langle s, a, r, s' \rangle$ to $\mathcal{D}$
7:     **if** $s'$ is terminal **then**
8:         $s \leftarrow$ env.reset()
9:     **end if**
10:     **for** $n$ in $\{1, \ldots, N\}$ **do**
11:         sample $\mathcal{B} = \{\langle s, a, r, s' \rangle\}$, compute $\nabla_w h(w)$
12:         $w \leftarrow \big(1 - (\alpha/\tilde{c})\big)w + (\alpha/\tilde{c})\theta - \alpha \nabla_w h(w)$
13:         numUpdates $\leftarrow$ numUpdates $+ 1$
14:         **if** numUpdates $\%$ *period* $= 0$ **then**
15:             $\theta \leftarrow w$
16:         **end if**
17:     **end for**
18: **until** convergence

---

## 10.2  Implementation Details

Table 1 and 2 show hyper-parameters, computing infrastructure, and libraries used for the experiments in this paper for all games tested. Our training and evaluation protocols and the hyper-parameter settings closely follow those of the Dopamine baseline. To report performance results, we measured the undiscounted sum of rewards obtained by the learned policy during evaluation.

| DQN hyper-parameters (shared) | |
|---|---|
| Replay buffer size | 200000 |
| Target update period | 8000 |
| Max steps per episode | 27000 |
| Evaluation frequency | 10000 |
| Batch size | 64 |
| Update period | 4 |
| Number of frame skip | 4 |
| Number of episodes to evaluate | 2 |
| Update horizon | 1 |
| $\epsilon$-greedy (training time) | 0.01 |
| $\epsilon$-greedy (evaluation time) | 0.001 |
| $\epsilon$-greedy decay period | 250000 |
| Burn-in period / Min replay size | 20000 |
| Learning rate | $10^{-4}$ |
| Discount factor ($\gamma$) | 0.99 |
| Total number of iterations | $3 \times 10^7$ |
| Sticky actions | True |
| Optimizer | Adam Kingma & Ba (2015) |
| Network architecture | Nature DQN network Mnih et al. (2015) |
| Random seeds | $\{0, 1, 2, 3, 4\}$ |
| **Rainbow hyper-parameters (shared)** | |
| Batch size | 64 |
| Other | Config file rainbow_aaai.gin from Dopamine |
| **DQN Pro and Rainbow Pro hyper-parameter** | |
| $\tilde{c}$ (DQN Pro) | 0.2 |
| $\tilde{c}$ (Rainbow Pro) | 0.05 |

Table 1: Hyper-parameters used for all methods for all 55 games of Atari-2600 benchmarks
. All results reported in our paper are averages over repeated runs initialized with each of the random seeds listed above and run for the listed number of episodes.

| Computing Infrastructure | |
|---|---|
| Machine Type | AWS EC2 - p2.16xlarge |
| GPU Family | Tesla K80 |
| CPU Family | Intel Xeon 2.30GHz |
| CUDA Version | 11.0 |
| NVIDIA-Driver | 450.80.02 |
| **Library Version** | |
| Python | 3.8.5 |
| Numpy | 1.20.1 |
| Gym | 0.18.0 |
| Pytorch | 1.8.0 |

Table 2: Computing infrastructure and software libraries used in all experiments in this paper.

## 10.3 Proofs

**Theorem 1.** *The Proximal Bellman Optimality Operator $\mathcal{T}^\star_{c,f}$ is a contraction with fixed point $v^\star$.*

We make two assumptions:

1. $f$ is smooth, or more specifically that its gradient is 1-Lipschitz: $||\nabla f(v_1) - \nabla f(v_2)|| \leq ||v_1 - v_2||\ \forall v_1, \forall v_2$ .
2. the value of the parameter $c$ is large, in particular $c > \frac{2}{1-\gamma}$ .

*Proof.* Both terms are convex and differentiable, therefore by setting the gradient to zero, we have:

$$\mathcal{T}^\star_{c,f}v = \mathcal{T}^\star v + \frac{1}{c}\big(\nabla f(v) - \nabla f(\mathcal{T}^\star_{c,f}v)\big),$$

We can then show:

$$
\begin{aligned}
||\mathcal{T}^\star_{c,f}v_1 - \mathcal{T}^\star_{c,f}v_2|| &= ||T^\star v_1 + \frac{1}{c}\big(\nabla f(v_1) - \nabla f(\mathcal{T}^\star_{c,f}v_1)\big) - T^\star v_2 - \frac{1}{c}\big(\nabla f(v_2) - \nabla f(\mathcal{T}^\star_{c,f}v_2)\big)|| \\
&\leq ||T^\star v_1 - T^\star v_2|| + \frac{1}{c}||\nabla f(v_1) - \nabla f(v_2)|| + \frac{1}{c}||\nabla f(\mathcal{T}^\star_{c,f}v_1) - \nabla f(\mathcal{T}^\star_{c,f}v_2)|| \\
&\quad \text{(first assumption)} \\
&\leq ||T^\star v_1 - T^\star v_2|| + \frac{1}{c}||\nabla f(v_1) - \nabla f(v_2)|| + \frac{1}{c}||\mathcal{T}^\star_{c,f}v_1 - \mathcal{T}^\star_{c,f}v_2||
\end{aligned}
$$

This implies:

$$
\begin{aligned}
\frac{c-1}{c}||\mathcal{T}^\star_{c,f}v_1 - \mathcal{T}^\star_{c,f}v_2|| &\leq ||T^\star v_1 - T^\star v_2|| + \frac{1}{c}||\nabla f(v_1) - \nabla f(v_2)|| \\
&\quad \text{(first assumption)} \\
&\leq ||T^\star v_1 - T^\star v_2|| + \frac{1}{c}||v_1 - v_2|| \\
&\leq \frac{\gamma c + 1}{c}||v_1 - v_2||
\end{aligned}
$$

Therefore,

$$||\mathcal{T}^\star_{c,f}v_1 - \mathcal{T}^\star_{c,f}v_2|| \leq \frac{\gamma c + 1}{c - 1}||v_1 - v_2||\,,$$

Allowing us to conclude that $\mathcal{T}^\star_{c,f}$ is a contraction (second assumption).

Further, to show that $v^\star$ is indeed the fixed point of $\mathcal{T}^\star_{c,f}$, notice from the original formulation:

$$\mathcal{T}^\star_{c,f}v := \arg\min_{v'} ||v' - \mathcal{T}^\star v||_2^2 + \frac{1}{c}D_f(v', v)\,,$$

that, at point $v^*$ setting $v' = v^\star$ jointly minimizes the first term, because $v^\star = \mathcal{T}^\star v^\star$ due to fixed-point defintion, and it also minimizes the second term because $D_f(v^\star, v^\star) = 0$ and that Bregman divergence is non-negative. Therefore, $\mathcal{T}^\star_{c,f}v^\star = v^\star$; $v^\star$ is the fixed-point of $\mathcal{T}^\star_{c,f}v^\star$. Since, $\mathcal{T}^\star_{c,f}$ is a contraction, this fixed point is unique. □

**Theorem 2.** *Consider the PMPI algorithm specified by:*

$$
\begin{aligned}
\pi_k &\leftarrow \mathcal{G}_{\epsilon'_k}v_{k-1}\,, & (4) \\
v_k &\leftarrow (\mathcal{T}^{\pi_k}_\beta)^n v_{k-1} + (1 - \beta)\epsilon_k\,. & (5)
\end{aligned}
$$

*Define the Bellman residual $b_k := v_k - \mathcal{T}^{\pi_{k+1}}v_k$, and error terms $x_k := (I - \gamma P^{\pi_k})\epsilon_k$ and $y_k := \gamma P^{\pi^*}\epsilon_k$. After $k$ steps:*

$$v^* - v^{\pi_k} = \underbrace{v^{\pi^*} - (\mathcal{T}^{\pi_{k+1}}_\beta)^n v_k}_{d_k} + \underbrace{(\mathcal{T}^{\pi_{k+1}}_\beta)^n v_k - v_{\pi_k}}_{s_k}$$

- *where $d_k \leq \gamma P^{\pi^*} d_{k-1} - \big((1-\beta)y_{k-1} + \beta b_{k-1}\big) + (1-\beta)\sum_{j=1}^{n-1}(\gamma P^{\pi_k})^j b_{k-1} + \epsilon'_k$*

- $s_k \leq \big((1-\beta)(\gamma P^{\pi_k})^n + \beta I\big)(I - \gamma P^{\pi_k})^{-1} b_{k-1}$

- $b_k \leq \big((1-\beta)(\gamma P^{\pi_k})^n + \beta I\big)b_{k-1} + (1-\beta)x_k + \epsilon'_{k+1}$

We make two assumptions:

1. we assume $\epsilon$ error in policy evaluation step, as already stated in equation (4).
2. we assume $\epsilon'$ error in policy greedification step $\pi_k \leftarrow \mathcal{G}_{\epsilon'_k} v_{k-1} \forall k$. This means $\forall \pi \; \mathcal{T}^\pi v_k - \mathcal{T}^{\pi_{k+1}} v_k \leq \epsilon'_{k+1}$. Note that this assumption is orthogonal to the thesis of our paper, but we kept it for generality.

*Proof.* Step 0: bound the Bellman residual: $b_k := v_k - \mathcal{T}^{\pi_{k+1}} v_k$ .

$$
\begin{aligned}
b_k &= v_k - \mathcal{T}^{\pi_{k+1}} v_k \\
&= v_k - \mathcal{T}^{\pi_k} v_k + \mathcal{T}^{\pi_k} v_k - \mathcal{T}^{\pi_{k+1}} v_k \\
&\quad \text{(from our assumption} \quad \forall \pi \; \mathcal{T}^\pi v_k - \mathcal{T}^{\pi_{k+1}} v_k \leq \epsilon'_{k+1}) \\
&\leq v_k - \mathcal{T}^{\pi_k} v_k + \epsilon'_{k+1} \\
&= v_k - (1-\beta)\epsilon_k - \mathcal{T}^{\pi_k} v_k + (1-\beta)\gamma P^{\pi_k}\epsilon_k + (1-\beta)\epsilon_k - (1-\beta)\gamma P^{\pi_k}\epsilon_k + \epsilon'_{k+1} \\
&\quad \Big(\text{from } \mathcal{T}^{\pi_k} v_k + (1-\beta)\gamma P^{\pi_k}\epsilon_k = \mathcal{T}^{\pi_k}(v_k - (1-\beta)\epsilon_k)\Big) \\
&= v_k - (1-\beta)\epsilon_k - \mathcal{T}^{\pi_k}(v_k - (1-\beta)\epsilon_k) + (1-\beta)\underbrace{(I - \gamma P_{\pi_k})\epsilon_k}_{x_k} + \epsilon'_{k+1} \\
&= v_k - (1-\beta)\epsilon_k - \mathcal{T}^{\pi_k}(v_k - (1-\beta)\epsilon_k) + (1-\beta)x_k + \epsilon'_{k+1} \\
&\quad \text{(from } v_k - (1-\beta)\epsilon_k = (\mathcal{T}_\beta^{\pi_k})^n v_{k-1}) \\
&= (1-\beta)(\mathcal{T}^{\pi_k})^n v_{k-1} + \beta v_{k-1} - \mathcal{T}^{\pi_k}\big((1-\beta)(\mathcal{T}^{\pi_k})^n v_{k-1} + \beta v_{k-1}\big) + (1-\beta)x_k + \epsilon'_{k+1} \\
&\quad \text{(from linearity of } \mathcal{T}^{\pi_k}) \\
&= (1-\beta)(\mathcal{T}^{\pi_k})^n v_{k-1} - \mathcal{T}^{\pi_k}\big((1-\beta)(\mathcal{T}^{\pi_k})^n v_{k-1}\big) + \beta\big(v_{k-1} - \mathcal{T}^{\pi_k} v_{k-1}\big) + (1-\beta)x_k + \epsilon'_{k+1} \\
&= (1-\beta)\Big((\mathcal{T}^{\pi_k})^n v_{k-1} - \mathcal{T}^{\pi_k}\big((T^{\pi_k})^n v_{k-1}\big)\Big) + \beta\big(v_{k-1} - \mathcal{T}^{\pi_k} v_{k-1}\big) + (1-\beta)x_k + \epsilon'_{k+1} \\
&= (1-\beta)\Big((\mathcal{T}^{\pi_k})^n v_{k-1} - (\mathcal{T}^{\pi_k})^n\big(\mathcal{T}^{\pi_k} v_{k-1}\big)\Big) + \beta\big(v_{k-1} - \mathcal{T}^{\pi_k} v_{k-1}\big) + (1-\beta)x_k + \epsilon'_{k+1} \\
&= (1-\beta)(\gamma P^{\pi_k})^n \big(\underbrace{v_{k-1} - \mathcal{T}^{\pi_k}(v_{k-1})}_{=b_{k-1}}\big) + \beta\big(\underbrace{v_{k-1} - \mathcal{T}^{\pi_k} v_{k-1}}_{=b_{k-1}}\big) + (1-\beta)x_k + \epsilon'_{k+1} \; ,
\end{aligned}
$$

allowing us to conclude:

$$
b_k = \big((1-\beta)(\gamma P^{\pi_k})^n + \beta I\big)b_{k-1} + (1-\beta)x_k + \epsilon'_{k+1} \; .
$$

Step 1: bound the distance to the optimal value: $d_{k+1} := v^* - (\mathcal{T}_\beta^{\pi_{k+1}})^n v_k$ .

$$
\begin{aligned}
d_{k+1} &= v^* - (\mathcal{T}_\beta^{\pi_{k+1}})^n v_k \\
&= \mathcal{T}^{\pi^*} v^* - \mathcal{T}^{\pi^*} v_k + \underbrace{\mathcal{T}^{\pi^*} v_k - \mathcal{T}^{\pi_{k+1}} v_k}_{\leq \epsilon'_{k+1}} + \underbrace{\mathcal{T}^{\pi_{k+1}} v_k - (\mathcal{T}_\beta^{\pi_{k+1}})^n v_k}_{=g_{k+1}} \\
&\leq \gamma P^{\pi^*}(v^* - v_k) + \epsilon'_{k+1} + g_{k+1} \\
&= \gamma P^{\pi^*}(v^* - v_k) + (1-\beta)\gamma P^{\pi^*}\epsilon_k - (1-\beta)\gamma P^{\pi^*}\epsilon_k + \epsilon'_{k+1} + g_{k+1} \\
&= \gamma P^{\pi^*}\Big(v^* - \big(v_k - (1-\beta)\epsilon_k\big)\Big) - (1-\beta)\underbrace{\gamma P^{\pi^*}\epsilon_k}_{y_k} + \epsilon'_{k+1} + g_{k+1} \\
&= \gamma P^{\pi^*}\big(\underbrace{v^* - (\mathcal{T}_\beta^{\pi_k})^n v_{k-1}}_{=d_k}\big) - (1-\beta)y_k + \epsilon'_{k+1} + g_{k+1} \\
&= \gamma P^{\pi^*} d_k - (1-\beta)y_k + \epsilon'_{k+1} + g_{k+1}
\end{aligned}
$$

Additionally we can bound $g_{k+1}$ as follows:

$$
\begin{aligned}
g_{k+1} &= \mathcal{T}^{\pi_{k+1}} v_k - (\mathcal{T}_\beta^{\pi_{k+1}})^n v_k \\
&= (1-\beta)\big(\mathcal{T}^{\pi_{k+1}} v_k - (\mathcal{T}^{\pi^{k+1}})^n v_k\big) + \beta(\mathcal{T}^{\pi_{k+1}} v_k - v_k) \\
&= (1-\beta)\sum_{j=1}^{n-1}(\gamma P^{\pi_{k+1}})^j b_k + \beta(-b_k)
\end{aligned}
$$

Allowing us to conclude that:

$$
d_{k+1} \le \gamma P^{\pi^*} d_k - \big((1-\beta)y_k + \beta b_k\big) + (1-\beta)\sum_{j=1}^{n-1}(\gamma P^{\pi_{k+1}})^j b_k + \epsilon'_{k+1}
$$

Step 2: bound the distance between the approximate value and the value of the policy: $s_k :=$ $(T_\beta^{\pi_k})^n v_{k-1} - v^{\pi_k}$ .

$$
\begin{aligned}
s_k &= (T_\beta^{\pi_k})^n v_{k-1} - v^{\pi_k} \\
&= (T_\beta^{\pi_k})^n v_{k-1} - (\mathcal{T}^{\pi_k})^\infty v_{k-1} \\
&= (1-\beta)(\mathcal{T}^{\pi_k})^n v_{k-1} + \beta v_{k-1} - (1-\beta)(\mathcal{T}^{\pi_k})^\infty v_{k-1} - \beta(\mathcal{T}^{\pi_k})^\infty v_{k-1} \\
&= (1-\beta)\big((\mathcal{T}^{\pi_k})^n v_{k-1} - (\mathcal{T}^{\pi_k})^\infty v_{k-1}\big) + \beta\big(v_{k-1} - (\mathcal{T}^{\pi_k})^\infty v_{k-1}\big) \\
&= (1-\beta)(\gamma P^{\pi_k})^n \big(v_{k-1} - (\mathcal{T}^{\pi_k})^\infty v_{k-1}\big) + \beta\big(v_{k-1} - (\mathcal{T}^{\pi_k})^\infty v_{k-1}\big) \\
&= \big((1-\beta)(\gamma P^{\pi_k})^n + \beta I\big)\big(v_{k-1} - (\mathcal{T}^{\pi_k})^\infty v_{k-1}\big) \\
&= \big((1-\beta)(\gamma P^{\pi_k})^n + \beta I\big)(I - \gamma P^{\pi_k})^{-1}\big(\underbrace{v_{k-1} - \mathcal{T}^{\pi_k} v_{k-1}}_{b_{k-1}}\big) .
\end{aligned}
$$

Allowing us to conclude that:

$$
s_k = \big((1-\beta)(\gamma P^{\pi_k})^n + \beta I\big)(I - \gamma P^{\pi_k})^{-1} b_{k-1} .
$$

$\square$

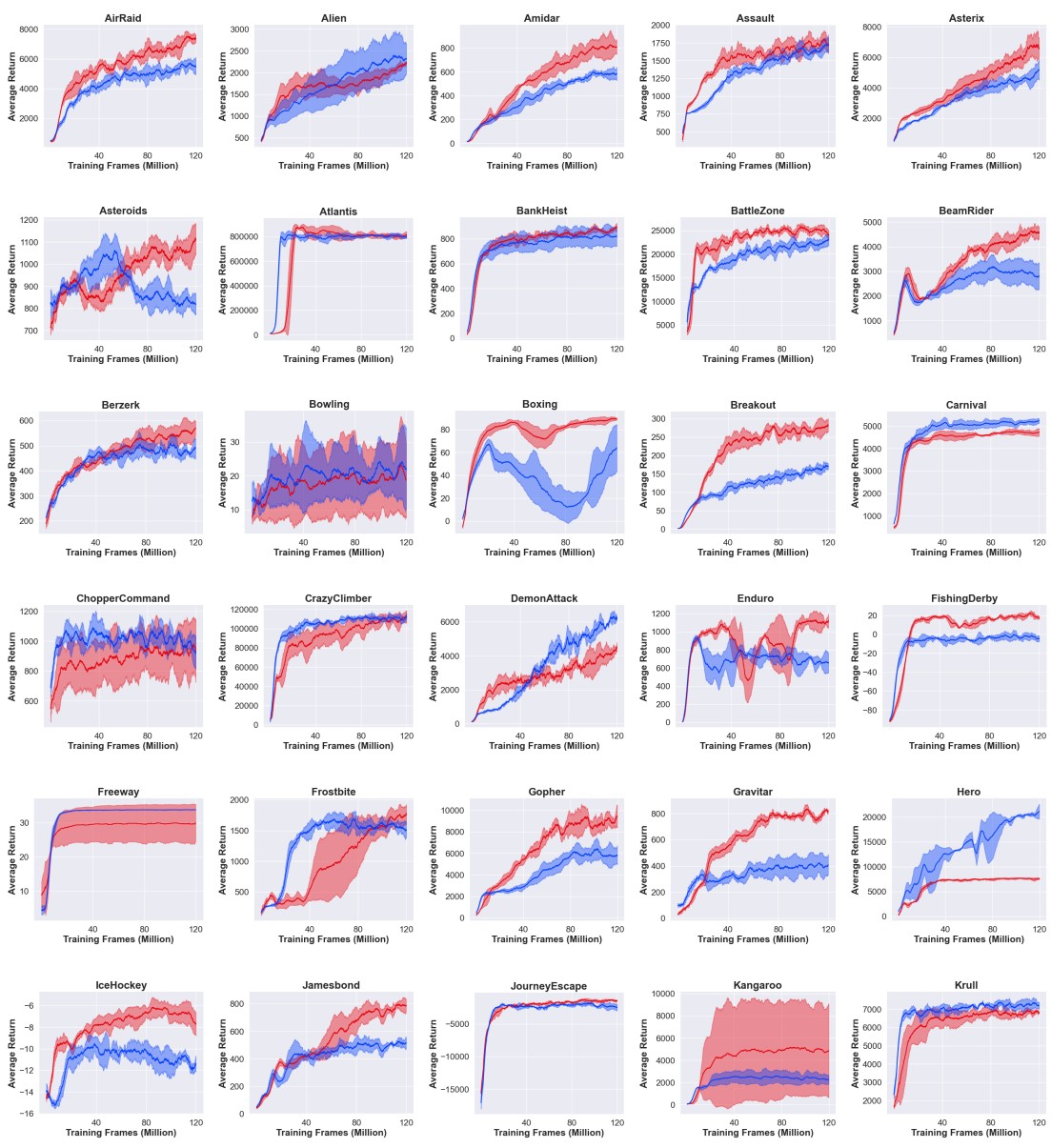

Figure 7: Comparison between DQN Pro (red) and DQN (blue) over 55 Atari games (Part I).

# 11 Learning curves

We present full learning curves of DQN, DQN Pro, Rainbow, and Rainbow Pro for the 55 Atari games. All results are averaged over 5 independent seeds.

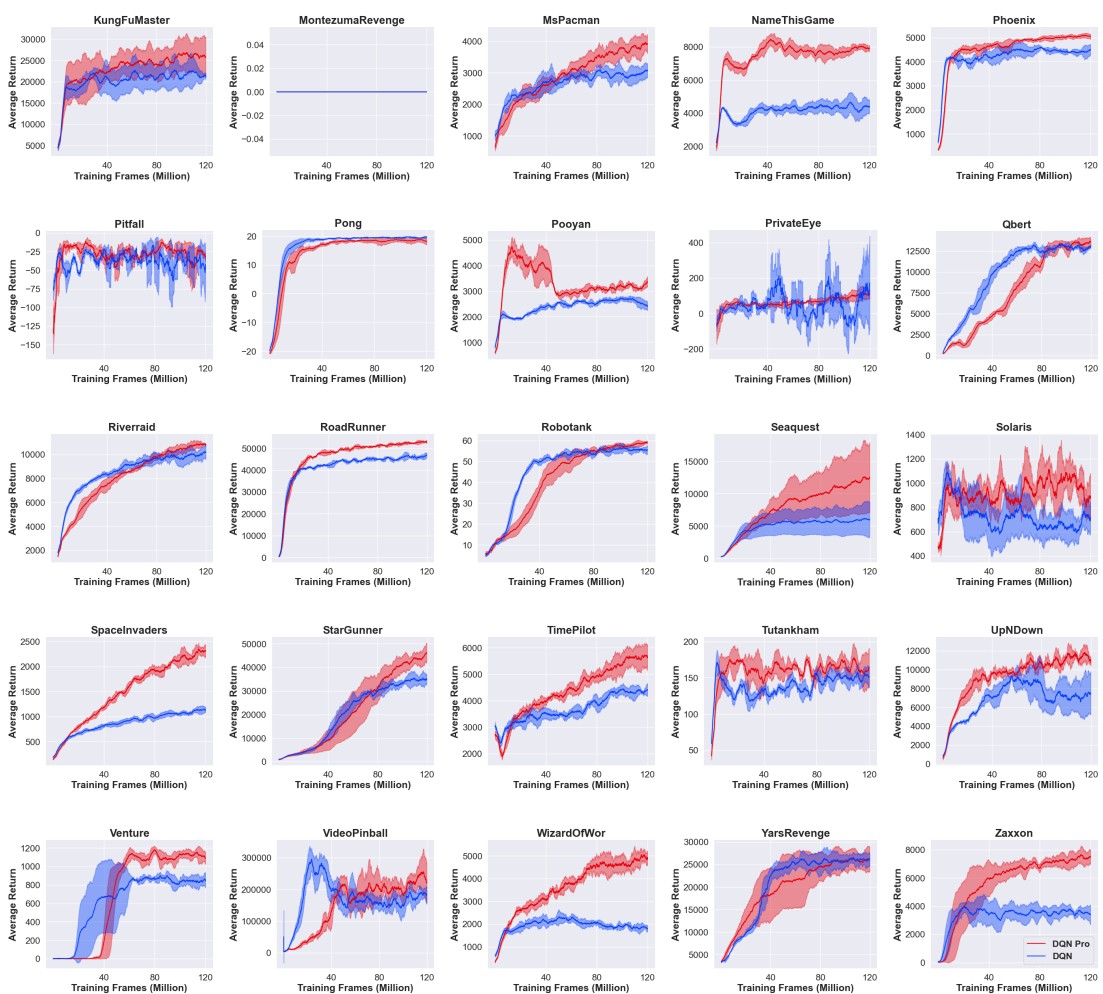

Figure 8: Comparison between DQN Pro (red) and DQN (blue) over 55 Atari games (Part II).

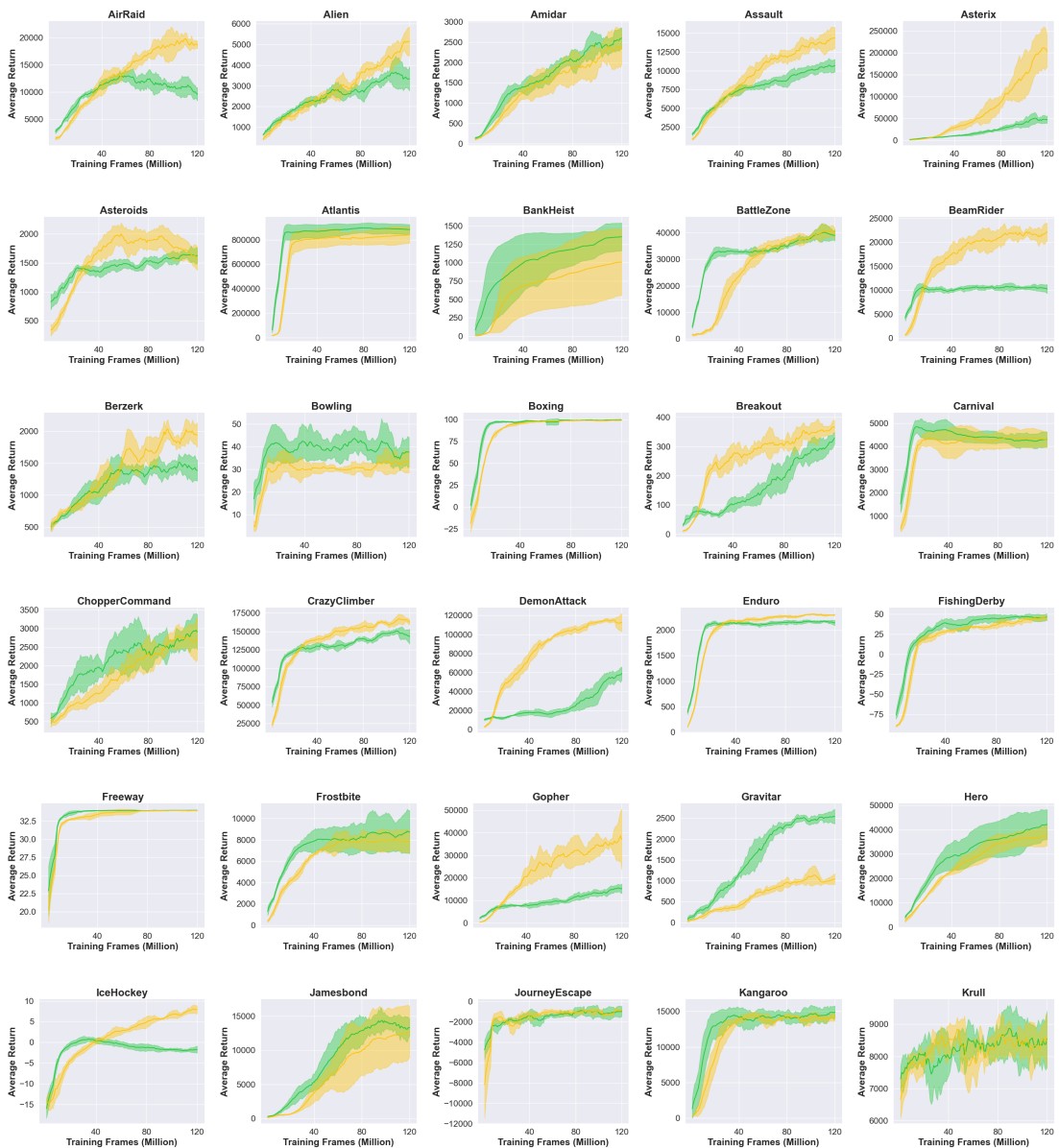

Figure 9: Comparison between Rainbow Pro (yellow) and Rainbow (green) over 55 Atari games (Part I).

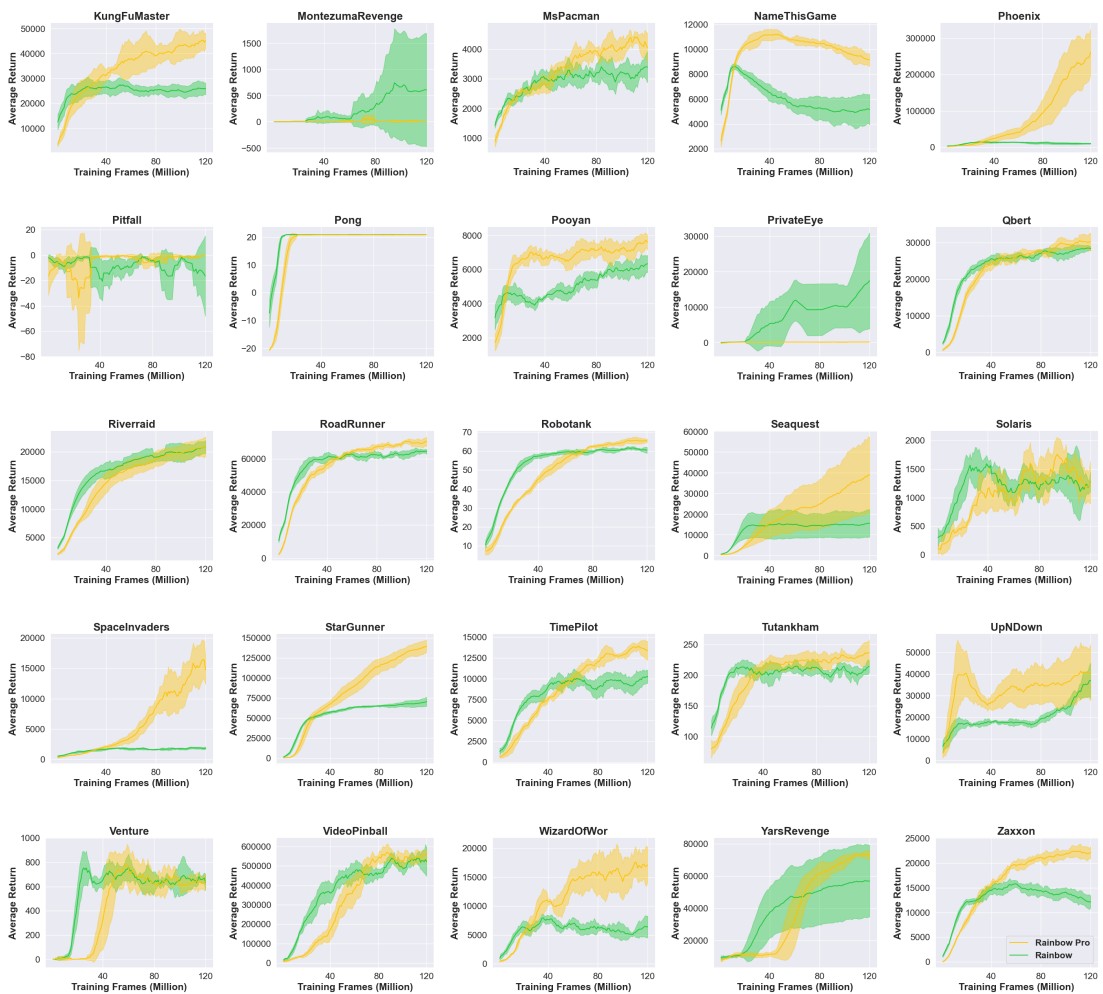

Figure 10: Comparison between Rainbow Pro (yellow) and Rainbow (green) over 55 Atari games (Part II).

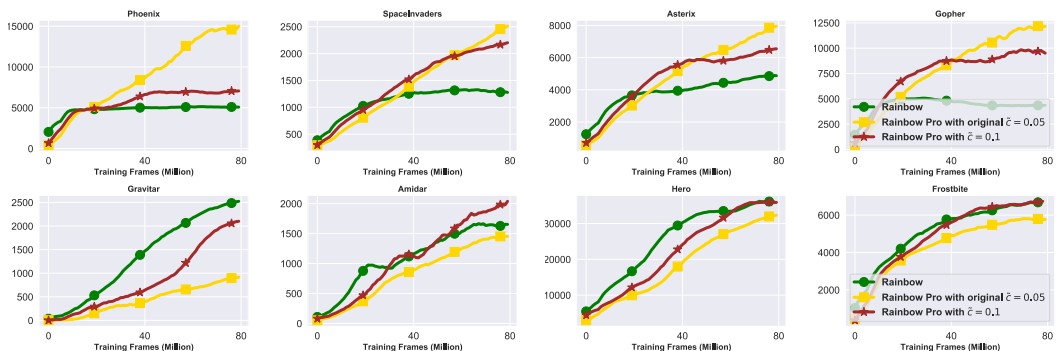

Figure 11: A study on games with the strongest (top) and weakest performance for Rainbow Pro with the original $\tilde{c} = 0.05$. Using a slightly less powerful proximal term (corresponding to larger $\tilde{c} = 0.1$) enables us to recover the downside (bottom) while still providing benefits on games that are more conducive to using the proximal updates (top).

## 12 Motivating Example for Adaptive $\tilde{c}$

In this section we specifically look at 4 domains in which Rainbow Pro did significantly better than the original Rainbow, as well 4 domains where Rainbow Pro is underperforming Rainbow. Note, again, that it is uncommon for Rainbow Pro with the original $\tilde{c}$ to underperform, but here we have a deeper dive into these cases for a better understanding.

From Figure 11, we observe that by using a slightly larger value of $\tilde{c}$, which slightly decreases the incentive for online-target proximity, we can recover from the downside, while still maintaining superior performance on games that are conducive to proximal updates. This suggest that, while using a fixed $\tilde{c}$ value is enough to obtain significant performance improvement, adaptively choosing $\tilde{c}$ would provide us with even more reliable improvements when performing proximal updates. In this context, a promising idea would be to hinge on the variance of our gradient updates when setting $\tilde{c}$. We leave this promising direction for future work.

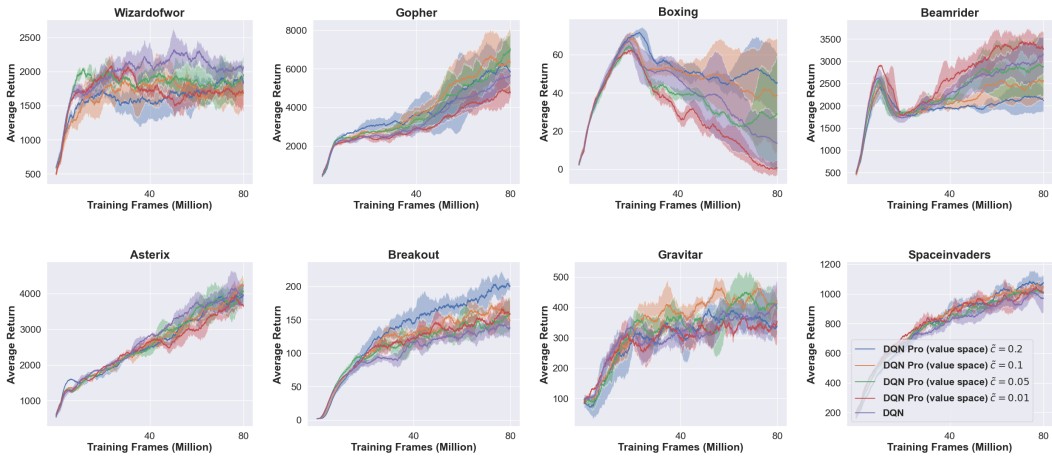

Figure 12: Performing proximal updates in the value space has a limited positive impact.

## 13 Proximal Updates in the Value Space

Our primary contribution was to show the usefulness of performing proximal updates in the parameter space. That said, we also implemented a version of proximal updates that operated in the value space. More specifically, in this case we updated the parameters of the online network as follows:

$$
w \leftarrow \widehat{\mathrm{E}}_{\langle s,a,r,s'\rangle} \left[ \left(r + \gamma \max_{a'} \widehat{Q}(s',a';\theta) - \widehat{Q}(s,a;w)\right)^2 \right] + \frac{1}{\tilde{c}} \widehat{\mathrm{E}}_{\langle s,a\rangle} \left[ \left(\widehat{Q}(s,a;w) - \widehat{Q}(s,a;\theta)\right)^2 \right].
$$

We conducted numerous experiments using variants of this idea (such as using separate replay buffer for each term, performing the update for all actions in buffered states, etc) but we generally found the value-space updates to be ineffective. As mentioned in the main paper, we believe this is because the parameter-space definition can enforce the proximity globally, while in the value space one can only hope to obtain proximity locally and on a batch of samples. To perform global updates we may need to compute the natural gradient, which typically requires matrix invasion Knight & Lerner (2018), and thus adding significant computational burden to the original algorithm. In comparison, the parameter-space version is effective, simple to implement, and capable of enforcing proximity globally due to the Lipschitz property of neural networks.