# OpenReview forum: "Faster Deep Reinforcement Learning with Slower Online Network"
_NeurIPS.cc/2022/Conference — NeurIPS 2022 Accept_

### Official Review · Reviewer_Gkox · 2022-07-04

**Rating:** 8
**Confidence:** 4
**Soundness:** 4 excellent
**Presentation:** 4 excellent
**Contribution:** 4 excellent

**Summary:**

The authors consider the class of deep reinforcement learning algorithms that are based on an online network that is updated at every interaction and a target network that is updated periodically. This technique, in general, ensures that the slower moving target network stabilizes the learning.

The paper proposes a type of regularization which keeps the online network in the proximity of the target network (this technique is referred to as the "slowing down" in the title of the paper).

The paper considers the DQN and Rainbow algorithms, and modifies them with the proposed approach. Experiments on the OpenAI Gym Atari game benchmarks show a significant improvement compared to the baseline algorithms. Experiments show that the performance improvement generated by technique depends on the level of noise in the reward function of the RL process.


**Questions:**

The paper provides sufficient information to make an informed opinion.


**Limitations:**

The paper adequately addresses the limitations of the work. The theoretical nature of the paper does not raise issues of negative societal impact.

**Strengths And Weaknesses:**

+ The paper is written very clearly.
+ Whenever feasible, theoretical motivations of the approach are given.
+ The paper clearly points out the parts where the current level of understanding does not allow for theoretical proofs.

---

> ### Author Response · Authors · 2022-08-02
> **Official Authors Response to Reviewer Gkox**
>
> We thank the reviewer for the positive assessment of our work, and for recognizing the impact and the effectiveness of proximal updates in value-function optimization in RL.

---

### Official Review · Reviewer_Djgd · 2022-07-10

**Rating:** 6
**Confidence:** 4
**Soundness:** 3 good
**Presentation:** 4 excellent
**Contribution:** 3 good

**Summary:**

In this paper, the authors propose to improve reinforcement learning robustness by essentially adding a regularization term to the Q updates, so that the weight of the online Q network will stay closer to the weight of the target Q network.
The authors provide both theoretical and empirical results and show that when the proposed method is added to DQN and Rainbow baselines, it improves performance quite significantly on Atari tasks.

**Questions:**

- My main question is: How is the proposed method fundamentally different from the alternatives? Why is it that it can achieve good performance improvement while others fail? (I also mentioned this in the previous section). In the current version of the paper, I don't quite find a very clear and comprehensive explanation for this.

**Limitations:**

I saw the authors indicate they have discussed the limitations of the work, but I did not seem to find where the discussion is, maybe I missed it? Would be great if you can point it out for me.

**Strengths And Weaknesses:**

Strengths:
- the simplicity of the method: as the authors discussed, the proposed method is quite simple and effective. And the simplicity brings a number of benefits.
- clean presentation and adequate technical details: the authors did a very good job presenting the method and also the many technical details, overall, the results seem to be reliable.
- interesting ablations: very important ablations to show that the effect of the proposed method is not the same as alternatives such as polyak averaging and learning rate decay.
- theoretical support: the theoretical results are good.
- significant results: although the proposed method did not improve on all tasks, but overall, the results show significant improvement in performance, and seems to be reliable results given the ablation provided and the discussion on hyperparameters.

Weaknesses:
- the novelty of the idea: the idea to make network weight stay close to the weight of an older version of the same network is not new. However, it would seem that this particular simple design will have a similar effect to changing polyak or learning rate decay, but the authors show that this is not the case and those ablations are very interesting. So probably should not be considered a major weakness.
- would love to see some more discussion on how exactly the proposed regularization can have superior effect compared to alternatives such as using polyak and smaller learning rates. Currently, there are empirical results that indicate the effects are very different, but why? For example, if we use polyak instead, will it perhaps... always produce a much larger shift in network weights or maybe the output Q values? I think it will be great if the authors can dig deeper and provide a hypothesis on what is the fundamental difference here that allowed the proposed method to achieve much stronger performance than alternative regularizations.
- would love to see some additional analysis on environments where the proposed method did not do well, in these cases, is it because the Q networks are now updated much slower (due to the fact that a consistent hyperparameter is selected for all environments)? Then for example, will increasing the learning rate in these environments recover the performance? Not a major point but would be good to see how that works.
- would be great if we can see results also for tasks with continuous action spaces, such as MuJoCo, where there seems to be a larger problem with bootstrapping. But I understand this might be significant extra work.


very minor typos:
- line 195 xCan

Summary:
Overall, I like this work, it is presented in a very clean fashion, with adequate details and good results. My current major concern is probably on the lack of explanation on how maintaining vicinity in the weight parameter space will give a very different effect compared to alternative methods? I am willing to increase my score if the rebuttal can address my concern.

---

> ### Author Response · Authors · 2022-08-02
> **Official Authors Response to Reviewer Djgd**
>
> - "would love to see some additional analysis on environments where the proposed method did not do well, in these cases, is it because the Q networks are now updated much slower (due to the fact that a consistent hyperparameter is selected for all environments)? Then for example, will increasing the learning rate in these environments recover the performance? Not a major point but would be good to see how that works."
>
> Per your suggestion, we ran another experiment (Section 11 and Figure 11 in the Appendix) where we dive deeper into some of the failure cases of our approach. While we were able to obtain significant overall improvements, we did see a few failure cases. This can occur in rare scenarios where RL updates are performed with minimal noise, and so using the proximal term results in suppressing the otherwise effective updates.
>
> As the reviewer predicted, we see that by slightly deemphasizing the proximal updates (slightly larger $\tilde c$), we not only successfully recover from the downside but still maintain the strong performance of Rainbow Pro in games that are more conducive to the proximal updates. We believe this result provides motivation for future work where we learn to adapt $\tilde c$ by using, for example, the variance of our gradient updates when performing value-function optimization. Using a fixed $\tilde c$ was quite simple and effective, which is why we used it here, but we do think that non-adaptive proximal updates are a limitation and we are eager to pursue the adaptive direction in future work. We appreciate your suggestion for carrying out this experiment. We also note that due to our time constraints for the rebuttal, we were able to collect up to 80 million frames as opposed to the usual 120 million frames, but we believe we should be able to draw a qualitatively similar conclusion upon completion of the experiment, and will add the complete 120 million frames to the final version.
>
> On this note, while we do not have a specific limitation section, we do mention our work's limitations across the submission and when applicable. For example, the distance function between the two value functions needs to be generated by a strictly convex function, or that our PMPI algorithm may not be a contraction for $n$>1.
>
> - "would love to see some more discussion on how exactly the proposed regularization can have superior effect compared to alternatives such as using polyak and smaller learning rates. Currently, there are empirical results that indicate the effects are very different, but why? For example, if we use polyak instead, will it perhaps... always produce a much larger shift in network weights or maybe the output Q values? I think it will be great if the authors can dig deeper and provide a hypothesis on what is the fundamental difference here that allowed the proposed method to achieve much stronger performance than alternative regularizations."
>
> More generally, we understand our proximal updates as an effective approach to addressing a trade-off between two desirable properties: a) accurately solving each iteration of value-function optimization in TD-like algorithms such as DQN and Rainbow and b) making fast progress to the fixed-point of the Bellman equation.
>
> Moreover, as demonstrated in Figures 5 and 6, step-size decay and Polyak averaging are not adequately addressing this trade-off. In particular, step-size decay prevents progress even when there is a clear and consistent gradient signal from the original RL objective function, which a vanishing step-size would overlook. This stands in contrast with the proximal updates that do not ignore consistent gradient signals. Moreover, using Polyak averaging results in rapidly-changing optimization problems, which unlike our proximal definition, results in instability during optimization in large and noisy environments.

---

> > ### Comment · Reviewer_Djgd · 2022-08-05
> > **Thank you for your response**
> >
> > I thank the authors for your response and for providing additional results. New results look good and you have made a number of fair comments. Based on what I saw so far I recommend acceptance of this work.

---

### Official Review · Reviewer_Pjai · 2022-07-11

**Rating:** 7
**Confidence:** 3
**Soundness:** 3 good
**Presentation:** 3 good
**Contribution:** 3 good

**Summary:**

The paper addresses the foundamental problem of Bellman operator in RL. The authors proposed a proximal Bellman operator that updates that incentivize the online network to remain in the vicinity of the target network. Theoretical analysis and toy task results suggested that the proximal Bellman Operator can accelerate convergence in the presence of high noise in value function approximation. Then the proximal value update is implemented on DQN and Rainbow agents and showed performance gain in many Atari games.

**Questions:**

Questions:
1. I understand $\alpha$ as the learning rate same as the orginal algorithm, is this correct?
2. While the experiments are based on Atari which are mostly deterministic environments,  I wonder whether the proposed PMPI works even better in noisy environments. May the author provide some insights?
3. Fig. 1, is $\epsilon$ equal to $\epsilon_k$ ?

Suggestions:

- Fig.2 and 5 are not that friendly to color-blind readers. Some efforts could be made to improved the plots.

**Strengths And Weaknesses:**

[Strength]
1. The work tackles a generally interested problem in deep RL about target value network.
2. The paper is basically well-written and easy to follow.
3. Both theoretical analysis and experimental results are well-presented
4. The proposed method is simple and effective.

[Weakness]
1. A new hyperparameter $\tilde{c}$ is introduced and need be to tunned to achieve best performance.
2. It is unclear that how can proposed PMPI generalize to continuous action case.

---

> ### Author Response · Authors · 2022-08-02
> **Official Authors Response to Reviewer Pjai**
>
> - "Fig.2 and 5 are not that friendly to color-blind readers. Some efforts could be made to improved the plots."
>
> Our sincere apologies for our visualization issues pertaining to color-blind readers. Per your suggestion, we have updated Figures 2, 4, 5, and 6. We used different marker shapes and/or used colorblind packages when possible. Please take a look at our revised submission and let us know if there are additional suggestions.
>
>
>
> - "I understand $\alpha$ as the learning rate same as the orginal algorithm, is this correct?"
>
> You are correct, $\alpha$ is indeed the learning rate for the original agent (DQN/Rainbow) and we did not change its value for our new Pro agents to ensure an easy and fair comparison. We also reiterate that while we do introduce a new hyper-parameter $\tilde c$, in our experience significant performance gain is possible with minimal tuning of this parameter. In particular, we used the same value of $\tilde c$ for all Atari games with no per-game tuning at all.
>
>
> - "While the experiments are based on Atari which are mostly deterministic environments, I wonder whether the proposed PMPI works even better in noisy environments. May the author provide some insights?"
>
> Regarding stochasticity in the problem, our Theorem 2 predicts that the Pro agents should theoretically perform better in presence of large noise. We tested and confirmed this hypothesis on the toy Policy Iteration experiments in Section 5.1.  We chose this toy setting to ensure there is no confounding factor. We observed an increasing advantage for using the Proximal Bellman Operator as we increase noise.
>
> We also note that our Atari environments are stochastic as we used sticky actions. More generally, existence of noise can stem from a variety factors such as policy stochasticity, function approximation, and sampling error. We agree that continuous actions could cause more errors, and therefore it would be intriguing to test this idea in the continuous setting. That said, in the continuous setting we generally directly learn a policy network due to problems that arise when computing the maximum Q value. We understand the addition of the policy network to be a crucial confounding factor, one that would make it challenging to carefully analyze the effects of using these slower updates. Thus, we left this question for future work.

---

> > ### Comment · Reviewer_Pjai · 2022-08-03
> > **Acknowledgement**
> >
> > Thanks for the reply. My concerns have been resolved, and thus I recommend an acceptance.

---

### Author Response · Authors · 2022-08-02
**General Authors Response**

We are delighted to learn that our submission has received unanimously positive reviews, and we thank our reviewers for their thoughtful feedback.

More specifically, we appreciate that Reviewer Pjai understands our paper to be tackling a generally interesting problem, we appreciate that Reviewers Pjai and Djgd see simplicity and adequate theory as a strength of our work, we appreciate that Reviewer Gkox acknowledges the significant performance improvements we obtain with this simple technique, and we are delighted to learn that all reviewers believe the paper to be well-written.

In our revision, we fixed a few typos and we addressed some of the issues that reviewers brought into our attention. We also note that we are committed to releasing an official Git repository for our paper to enable the community to fully reproduce our results.

We use official comments to respond to each review individually.

---

### Meta-Review · Area_Chair_U82R · 2022-08-23

**Recommendation:** Accept
**Confidence:** Certain

**Metareview:**

This paper proposes a a simple method to improve the robustness and sample-efficiency of value-based deep RL methods. The idea is to regularize parameters not to deviate much from target-network parameters. The paper also provides theoretical results showing its convergence property and justifying why the proposed method can accelerate convergence. The results across all Atari games are strong. All of the reviewers appreciated the simplicity and the effectiveness of the method. Some of the minor concerns were addressed during the rebuttal period. Thus, I recommend accepting this paper.

**Award:**

No

---

### Decision · Program_Chairs · 2022-09-14

Accept